# Mesostructured Fibrils Exfoliated in Deep Eutectic Solvent as Building Blocks of Collagen Membranes

**DOI:** 10.3390/polym15194008

**Published:** 2023-10-06

**Authors:** Ying Pei, Wei Li, Lu Wang, Jing Cui, Lu Li, Shengjie Ling, Keyong Tang, Huafeng Tian

**Affiliations:** 1Key Laboratory of Auxiliary Chemistry and Technology for Chemical Industry, Ministry of Education, Shaanxi University of Science and Technology, Xi’an 710021, China; lilu@sust.edu.cn; 2Key Laboratory of Processing and Quality Evaluation Technology of Green Plastics of China National Light Industry Council, Beijing Technology and Business University, Beijing 100048, China; 3College of Materials Science and Engineering, Zhengzhou University, Zhengzhou 450001, China; lw15565152810@126.com (W.L.); 18839150063@163.com (L.W.); kytangzzu@126.com (K.T.); 4School of Physical Science and Technology, Shanghai Tech University, Shanghai 201210, China; cuijing@shanghaitech.edu.cn

**Keywords:** collagen, bovine Achilles tendon, fibrils, exfoliation, deep eutectic solvent

## Abstract

The mesoscale components of collagen (nanofibrils, fibrils, and fiber bundles) are well organized in native tissues, resulting in superior properties and diverse functions. In this paper, we present a simple and controlled liquid exfoliation method to directly extract medium-sized collagen fibers ranging from 102 to 159 nm in diameter from bovine Achilles tendon using urea/hydrochloric acid and a deep eutectic solvent (DES). In situ observations under polarized light microscopy (POM) and molecular dynamics simulations revealed the effects of urea and GuHCl on tendon collagen. FTIR study results confirmed that these fibrils retained the typical structural characteristics of type I collagen. These shed collagen fibrils were then used as building blocks to create independent collagen membranes with good and stable mechanical properties, excellent barrier properties, and cell compatibility. A new method for collagen processing is provided in this work by using DES-assisted liquid exfoliation for constructing robust collagen membranes with mesoscale collagen fibrils as building blocks.

## 1. Introduction

Collagen, as an essential component in the extracellular matrix, is the most widely distributed in most connective tissues of mammals, such as bone, tendon, skin, cartilage, cornea, and blood vessels [1]. In the collagen family, type I collagen has the most abundant content, accounting for 90% weight of the total collagen and 20% weight of the total protein [2]. Each year, a massive amount of collagen-containing waste and by-products are generated from the leather and food industries, providing a huge opportunity for the reuse of natural collagen. It has been estimated that a tanning company in the Balkans spends up to USD 400,000 annually on the transportation and landfill of collagen-containing waste [3]. It has been estimated that the total amount of collagenous waste generated from slaughterhouse disposals in India is over 2.1 million tons each year [4]. Moreover, the remarkable performance features of collagen, such as low antigenicity, good biocompatibility, and biodegradability, have generated interest in its high-value-added utilization in tissue engineering and intelligent healthcare [5,6,7,8]. Therefore, the recycling and utilization of collagen waste have been highlighted to reduce the environmental burden and promote the sustainable development of the industrial economy. 

In routine processing, collagen is extracted and dissolved with the assistance of acid [9], alkaline [10], salt [11], enzyme and ionic liquids [12], etc., involving tedious procedures and special storage requirements, which limit the large-scale manufacturing of collagen-based materials. The resulting collagen solutions were applied for modeling materials with diverse formats through chemical treatments [13,14] and physical assemblies [12,15]. However, the aggregated structure of natural collagen is destroyed or lost in the dissolution process, leading to the loss of its inherent performance advantages at the mesoscale. Therefore, collagen materials derived from collagen solutions through molecular-level designing and processing have the disadvantages of weak mechanical properties, low thermal stability, and poor structural stability. For instance, the elastic modulus of collagen fibrils from bovine tendons reaches up to 5 GPa [16], while the modulus of collagen nanofibers prepared by electrospinning a collagen solution is only 52.3 MPa [17]. The strategies of chemical modification, cross-linking treatment, and blending with other substances have been reported to enhance the interactions among collagen molecules and improve the material’s structural stability and mechanical properties. Some arising issues include safety concerns related to the chemical cross-linking agents used, the loss of collagen characteristics due to chemical modification, and the compatibility of the blending system. Moreover, it is challenging to construct a well-organized structure since collagen’s self-assembly behaviors are affected by complicated factors such as terminal peptide type, sequence length, temperature, pH value, ionic strength, medium type, etc. Therefore, new methods of collagen processing need to be developed.

Type I collagen consists of sophisticated hierarchical structures from the nanoscale to the macroscale. Microfibrils, fibers, and fiber bundles are natural collagen assemblies (CAs), endowing biological tissues with exceptionally strong, extensible, and tough mechanical performance [18,19]. A variety of approaches have been developed to extract CAs to retain the inherent architectures to provide building blocks for collagen materials with high performance [20,21,22]. These approaches have included physical treatment (grinder [23], homogenizer [24,25], and blender [16,21,26,27] treatments) and solvent assistance (acid [28,29,30] and acid–enzyme [31]). However, obtaining CAs with controllable structures is challenging due to the complex architectures of collagen in biological tissues. Starting with disassembling the noncovalent bond interactions that stabilize the collagen structure, our group reported a controllable method to exfoliate collagen fibrils from bovine Achilles tendons in a sodium hydroxide (NaOH)/aqueous urea system using sonication treatment. The successful stripping of collagen fibrils is due to the intensive hydrogen bond interactions between urea and collagen in alkaline conditions at lower temperatures [20]. We were inspired by the liquid exfoliation of nanofibrils in deep eutectic solvents (DESs) [32,33,34,35,36,37,38,39,40] and green and tailorable solvents for the treatment of biomass [33,41,42,43]. Urea/GuHCl is a promising powerful protein denaturant for the direct exfoliation of silk nanofibers by disrupting noncovalent bonds and decreasing the hydrophobic interactions in proteins [36,43]. In this work, we elaborate on a facile and controllable technique for the exfoliation of mesostructured collagen fibers in urea/GuHCl, providing a method for collagen processing. The resulting collagen fibers are building blocks of free-standing membranes with outstanding mechanical properties, biocompatibility, and optical properties. 

## 2. Experiments

### 2.1. Materials and Chemicals

Bovine Achilles tendons were obtained from a slaughterhouse nearby. Sodium hydroxide (NaOH, ≥99.7%), urea (N_2_H_4_CO, ≥99%), guanidine hydrochloride (CH_5_N_3_·HCl, ≥99%), acetic acid (C_2_H_4_O_2_, ≥99%), sodium dodecyl sulfate (C_12_H_25_O_4_NaS, ≥99.5%), and tris(hydroxymethyl)aminomethane hydrochloride (Tris-HCl, ≥99%) were purchased from Macklin Biochemical Co., Ltd., Shanghai, China. A 2% phosphotungstic acid negative stain solution was brought from Solarbio Science and Technology Co., Ltd., Beijing, China. Mouse connective tissue fibroblasts L-929 (China Center for Type Culture Collection) were routinely cultured using Dulbecco’s modified Eagle’s medium and 10% fetal bovine serum (CHI Scientific, Inc., Maynard, MA, USA). All the chemicals were used as received without further purification. Milli-Q water (Merck Millipore, Darmstadt, Germany, electrical conductivity ≈ 18.1 MΩ cm^−1^) was utilized in all the experiments.

### 2.2. Exfoliation of Collagen Fibrils

Using a physical tool, beef tendons were removed from fat and noncollagen components and cut into small pieces of about 1 cm^3^. In decellularization treatment, the beef tendons were then treated with 1% sodium dodecyl sulfate along with 1 mM Tris−HCl and 0.1 mM EDTA at 4 °C for 20~24 h. Thereafter, the beef tendon pieces were soaked in 0.2 wt% sodium hydroxide solution and fully swollen for 12 h and then rinsed with distilled water. Beef tendon slurry was obtained by treating swollen cuts in a high-speed mixer and then lyophilized to obtain pre-treated beef tendons. Appendix A shows the experimental process. Specifically, a DES was prepared by mixing GuHCl and urea in the molar ratio of 2:1 at 70 °C until a clear system. The pre-treated beef tendon was immersed in the DES for 10 min to obtain a gel-like mixture. This mixture was washed with distilled water to remove the DES. The DES-treated tendons were redispersed in an acetic acid solution (0.01 *v*/*v*) at a weight ratio of 1:100 and treated with a high-pressure homogenizer at 45 MPa for 5, 20, and 30 times to generate collagen fibrils (CFs). Samples were denoted as CFs-5, CFs-20 and CFs-30. To quantify the yield of CFs, filters were used during CF dispersion using vacuum filtration on a commercial polycarbonate membrane by pre-weighting. The CF-loaded membrane was dried and weighted. The CFs’ yield was determined by measuring the difference in filtration membranes. 

### 2.3. Preparation of Collagen Membranes

The CFs were dispersed in a 0.03 *v*/*v* acetic acid solution for uniform dispersion. The CF membrane was prepared using vacuum filtration during CFs’ acetic acid dispersion with polycarbonate membranes with a pore size of 0.2 μm and a diameter of 47 mm for 8 h. These membranes were denoted according to the times of the performed homogenization for CFs, such as M-5, M-20, and M-30. Membranes made from unhomogenized fibers are labeled as M-control.

### 2.4. Characterization

The exfoliation process for mesoscale collagen fibrils was observed in situ using polarized light microscopy (POM, DM4P, Beijing Groupca Technology Co., Ltd., Beijing, China). Briefly, fiber bundles (~300 μm) from pre-treated bovine tendons were loaded on a glass slide. A drop of DES and distilled water were dripped onto fiber bundles and observed under POM. The morphology of collagen fibrils was observed on ultra-depth optical microscopy (VHX-6000, Keyence Corporation, Osaka, Japan). The collagen fibrils and collagen membranes were sputter-coated with gold and observed via scanning electron microscopy (QuantaFEG250, FEI, Hillsboro, OR, USA) at an acceleration voltage of 20 kV. The diameter and length of collagen fibrils were measured by analyzing SEM images with ImageJ software (version, 1.8.0). FTIR tests of collagen fibrils were carried out on a Jasco FTIR-6200 spectrometer in the range of 800–4000 cm^−1^ with a resolution of 4 cm^−1^.

### 2.5. Computer Simulations

Molecular dynamics (MD) simulations were performed using a visual Gromacs-software, (version, 2020.4) and the GROMOS96/53a6 force field. The initial type I collagen segments were obtained from the Protein Data Bank (2MQS) [44]. The simple energy optimization of collagen molecules was carried out to form a stable triple-helix architecture. After energy minimization, collagen molecules were placed in a periodic box with 1 × 10^4^ DES molecules based on Lennard–Jones–Coulomb atom–atom potentials. The systems were pre-minimized to remove nonspecific contacts and steric hindrances. MD calculations were carried out by performing 5000 steps of the steepest descent algorithm (1 fs per step). Minimization was stopped until the maximum force <1000 kJ mol^−1^ nm^−1^, in order to remove physically unfavorable molecular configurations. The constant volume (NVT) ensemble (343K) and constant pressure (NPT) ensemble (1.0 bar) were carried out in MD simulations. Periodic boundary simulations based on the particle mesh Ewald (PME) method were performed by using the NPT method with a cutoff at 1 nm at constant temperature and pressure. The long-range coulomb forces were calculated using the PME method by using a 1.0 nm cutoff. The root-mean-square deviation (RMSD) and relative basis mean square fluctuation (RMSF) values were determined with the GROMACS tools. 

### 2.6. Tensile Test

Rectangular membrane samples (25 mm × 7 mm) with an average thickness of 0.3 mm were used for all mechanical tests. Tensile tests were carried out at room temperature using a universal tensile analyzer (TA XT Plus, Stable Micro Systems, Godalming, Surrey, UK) at 25 °C and relative humidity (RH) of 23% at a tensile speed of 0.1 mm/s. The experimental data were determined as the average of the test data of five samples, and typical stress–strain curves were chosen for analysis.

### 2.7. Optical Test

A UV–Vis spectrometer (TU-1950) was utilized to measure the optical transmission of collagen membranes in the range of 400–800 nm. The transmission data of the dry and hydrated collagen membranes were collected.

### 2.8. Thermal Stability Test

The thermal analysis of membranes was performed by using a TG-DSC Simultaneous Thermal Analyzer (Mettler Toledo, Greifensee, Switzerland). For this test, 10 mg membrane samples were sealed in aluminum crucibles under a N_2_ atmosphere with a flow rate of 20 mL/min (heating rate of 10 °C/min).

### 2.9. Water Vapor Permeability

Water vapor permeability (WVP) tests of collagen membranes were carried out on a Water vapor transmittance tester (3/33MA, MOCON, Brooklyn Park, MN, USA) according to ASTM standard E96. The membranes were placed onto the test cups loaded with 20 mL of deionized water. The differences in relative humidity (∆RH) across the membranes and the test temperature were controlled with the tester at 70 ± 1% and 25 °C, respectively. The following equation was used to calculate the WVP values:$$\mathrm{WVP}=\frac{\mathrm{WVTR}\times T}{P_{0}\times ∆\mathrm{RH}\times A}$$
where the water vapor transmission rate (WVTR) is obtained from the slope of the plot of weight loss versus the testing time, and the saturated water vapor pressure *P*_0_ at 25 °C is 3160 Pa. The thickness (*T*) and area (*A*) of membranes were 0.3 mm and 100 cm^2^, respectively.

### 2.10. Oxygen Permeability Test

A certain amount of soybean oil was placed in a vessel with 2 cm^2^ leaky caps. The collagen membranes were cut into proper areas and installed onto the vessels to cover the leaky areas of caps for 48 h at room temperature, 50 RH%. The peroxidation values (POVs) of soybean oil samples in vessels were determined using a 0.002 mol/L sodium thiosulfate standard solution according to China’s National Standard of Food Safety (GB5009.227-2016). POV determinations of vessels without membrane cover were used as a control. The experimental data were determined as the average of the test data of three samples. The initial POV of oil was about 3.0 mmol/kg.

### 2.11. In Vitro Cells Culturing

Mouse fibroblast L929 cells (China Center for Type Culture Collection) were cultured in Dulbecco’s modified Eagle’s medium (DMEM, Gibco, Thermo Fisher Scientific Inc., Waltham, MA, USA) with 10% fetal bovine serum and 1% penicillin–streptomycin (Gibco). The membranes were sterilized using ethanol and ultraviolet radiation irradiation. L929 cell suspension (2 × 10^5^ cells/mL) was carried out by seeding the cells on the surface of membranes and developing culture in an incubator at 37 °C with 5% CO_2_. The metabolic activity was determined using 3-(4, 5)-dimethylthiahiazo (-z-y1)-3, 5-diphenytetrazoliumromide (MTT) assay. The absorbance was recorded at 570 nm on a microplate reader (Multiskan GO, Thermo, Waltham, MA, USA). The fold change in metabolic activity was calculated for days 4, 7, and 10 and compared with day 1.

### 2.12. Data Analysis

The size of the collagen fibrils was analyzed by measuring 20 independent locations in SEM images. Three parallel experiments were conducted to obtain the calculated average of the yield. Tensile tests of membranes were carried out on five sample replicates, and the typical stress–strain curves were plotted. Transmittance tests of membranes were carried out by measuring three independent locations in one membrane sample, and the typical stress–strain curves were plotted. The MVP and POV of each membrane sample were measured three times, and the mean value was obtained. The analysis of cells’ metabolic activity was performed on six independent replicates for each membrane sample. Origin Software (version 9.1) was used to determine statistical significance, and the levels were set as * *p* < 0.1 and ** *p* < 0.01.

## 3. Results and Discussion

### 3.1. Structure of Mesoscale Collagen Fibrils

In this work, collagen fibrils were obtained using a scalable liquid exfoliation method in urea/GuHCl, as shown in Figure 1a. The pre-treated bovine tendon was added into urea/GuHCl for liquid exfoliation, followed by homogenization, to obtain collagen fibrils with smaller sizes. In an adult bovine tendon tissue, a certain number of individual collagen fibrils with a 100–300 nm diameter were arranged into a fiber bundle along the fibrils’ axial structure (Figure 1b). After urea/GuHCl treatment, some collagen fibrils were split from the fiber bundle (Figure 1c). The effects of the weight ratio (collagen–GuHCl *w*:*w*) and contacting time on the yield of collagen fibrils are shown in Appendix A. At 10 min, the fibrils of highest yield 89% were harvested (Appendix A). The yield decreased with an increase in the contacting time, which is attributed to the partial dissolution of collagen in the DES. The dosage of GuHCl also greatly influenced the yield (Appendix A). The highest yield was obtained when the weight ratio of collagen and GuHCl was 1:40. GuHCl of low dosage could not form a full interaction with collagen, while too high dosage further destroyed collagen, resulting in reduced yield. The optimum exfoliation of collagen occurred when the weight ratio of collagen and GuHCl was 1:40 with a contacting time of 10 min. The exfoliated collagen fibrils presented the typical axial D-period repeats (Figure 1d) of type I collagen. By performing homogenization, collagen fibrils of smaller sizes were harvested (Figure 1e–g). After homogenization, collagen fibrils still retained the D-band structure (Appendix A). The size of the fibrils was measured and analyzed using SEM images, which are presented in Appendix A. With the increase in the number of homogenization procedures performed, the length of the obtained collagen fibrils gradually decreased. However, the difference in length was relatively small. This is attributed to the way the shear force of the homogenizer works. Therefore, the size of collagen fibrils could be changed by adjusting the number of homogenization operations conducted. 

### 3.2. Analysis of Exfoliation Process

The exfoliation process for mesoscale collagen fibrils was observed in situ using POM. Fiber bundles obtained from pre-treated bovine tendons (200–300 μm) were selected for observation. In Figure 2, bright zones along the fiber axis are on fiber bundles, which are crystalline collagen regions. Many tightly bound collagen aggregate structures are present in these crystalline regions. After being immersed in urea/GuHCl, slight swelling appeared on bundles. The partially disappeared crystalline region on the bundle suggested the collapse of fibril packing. Our previous work has shown that the exfoliation of collagen fibrils is attributed to the hydrogen bonds’ interaction between urea and collagen. The entropy of randomly coiled collagen increases because of urea penetration into crystalline regions [45]. Guanidine hydrochloride has the ability to disrupt the noncovalent bonds and weaken the strength of hydrophobic interactions in proteins [36]. It has been reported that GuHCl can be an excellent candidate to exfoliate silk nanofibers by breaking hydrogen and hydrophobic bonds. To further analyze the effect of temperature, we observed the collagen fiber bundles in both dry and humid heat states. As shown in Figure 2, regardless of the shape and crystal area of the fiber, the fiber bundle does not change in air or water. This shows that the swelling of the fibers and the destruction of the grain areas are not directly dependent on temperature but on the action of the DES and fiber bundles. Compared with the previous NaOH/urea, a strongly alkaline system, urea/GuHCl provides a milder condition for exfoliation. Moreover, the exfoliation of collagen in the NaOH/urea system lasted several hours due to multiple freezing and thawing cycles treatments, while the exfoliation of collagen in the urea/GuHCl system required treatment for only 10 min. 

MD simulations were performed to analyze the changes in the structure of collagen chains in urea/GuHCl (Figure 3). We focused on the interactions of hydrogen bonds among urea, GuHCl, and collagen chains. Collagen molecules were generated with three α helix chains, comprising two α1 (I) chains and one α2 (II) chain. Hydrogen bond interactions stabilized the triple helix [46]. Notably, the dominant interactions between urea and collagen were observed by comparing the hydrogen bond numbers in Figure 3a, suggesting intensive hydrogen bond interactions. Hydrogen bond donors or acceptors are provided by lone pairs of electrons on the O atom of urea, which form hydrogen bonds with polar and nonpolar groups of the peptide. Moreover, coulomb force interactions among urea, GuHCl, and collagen chains are revealed in Figure 3b. More coulomb force interactions occurred between GuHCl and the collagen chain than between urea and the collagen chain. The change in RMSD reveals the stability of the dynamic protein system. As shown in Appendix A, the RMSD of collagen macromolecular reaches a rapid equilibrium with significant fluctuations, suggesting good stability of the structure in the urea/GuHCl system. The results show that urea and GuHCl can form nonchemical interactions with collagen with minimal impact on collagen configuration. FTIR spectra were generated to analyze the changes in the collagen structure further, as shown in Figure 3c. The region of 1650−1668 cm^−1^ is mainly associated with the C=O stretching vibrations of the amide I bond in the collagen chain [47], which is a sensitive marker of the peptide conformation. The amide II region existing in 1530−1555 cm^−1^ is attributed to both −N−H deformation and C−N stretching [47]. Amide III is also a sensitive marker of collagen secondary structure, which is in the range of 1310−1175 cm^−1^, attributed to the C–N stretching and N–H bending vibrations [48]. The above characteristics are reflected in the spectra of all samples, indicating that the secondary structure of native collagen was retained during exfoliation.

### 3.3. Properties of Collagen Membranes

The free-standing collagen membranes (Figure 4a, thickness ∼30 μm) exhibited excellent transparency and flexibility. The membrane thickness could be adjusted by changing collagen suspension during filtration (Appendix A). The dry collagen membrane exhibited a compact layered structure formed by the accumulation of collagen fibrils during filtration (Figure 4b,c). The optical transmittance of the membrane samples is shown in Figure 4d. An obvious UV resistivity appears in the 200–240 nm wavelength due to the weak absorption of disulfide bonds formed by two cysteines in the collagen chain around 240 nm. The optical transmittance of collagen membranes is enhanced with the increase in homogenization times, resulting in the consequent reduction in fiber size. Smaller-size collagen fibrils are densely packed, resulting in a decrease in the light scattering phenomenon. As shown in Figure 4e and Appendix A, for dry membranes, the dry collagen membranes present a brittle fracture. The decreased size induced the enhanced rigidity of collagen fibrils, increasing the homogenization times and resulting in higher stress and lower strain at break under tensile force. The maximum stress of dry collagen membranes reached 104.99 MPa, and the nonhomogenized fibril-formed membrane produced a maximum strain of 30%. Smaller-size fibers formed more intensive interactions than larger-size ones, resulting in improved collagen membrane performance when act as building blocks. However, the fibrils with a low aspect ratio had poor resistance to deformation, which led to increased brittleness of the collagen membrane. Compared with other reported collagen-based membranes (Appendix A) [49,50,51,52], collagen membranes presented excellent mechanical properties in this work. The membrane with collagen fibrils as building blocks showed superior performance compared with those formed from molecular assembly in collagen solutions. Figure 4f and Appendix A show the tensile mechanical properties of the hydrated membranes. The tensile stress at the break was 0.02–0.4 MPa, and the strain was 17−97%. These results indicate that collagen membranes’ mechanical properties depend on collagen fibrils’ size. It is worth noting that the stress–strain curves of hydrated membranes have a “J-shape”. At lower tensions, the curves have a peak region (I), where the fibrils in the membrane straighten and align along the tension direction. Subsequently, the oriented fibrils deform in the nonlinear regions of the curves (stage II) and begin to slide against each other, and the stress–strain behavior is mainly determined by the stretching of the elastic fibrils. The plastic region in which highly oriented fibrils are further deformed is associated with a significant increase in tensile strength. In stage III, the increase in linear area is attributed to the straightening of kinks among the collagen macromolecules, resulting in the reduced disruption of lateral molecular packing in the fibrils. This stretching is also related to the displacement of neighboring collagen molecules. This similar curve shape has been observed in many biological tissues, including ligaments, skin, and blood vessels [53,54,55]. Therefore, the present collagen membranes have the potential to be used in tissue engineering. The barrier of collagen membranes to water vapor and oxygen is shown in Figure 4g,h. The moisture content of the collagen membranes decreased with the decrease in the size of fibrils. Compared with the M-control, the M-30 with a smaller fibril size improved the water vapor barrier by 14%. The smaller fibrils formed a dense membrane structure during vacuum filtration, effectively blocking water molecules. Moreover, the same trend was observed in the oxygen barrier of collagen membranes. Due to the dense fibrous membrane structure formed by mesoscopic units, the collagen membranes effectively reduced the POV value of the oils. Therefore, given the excellent barrier properties of collagen membranes, they can be used as food packaging materials.

L929 fibroblasts were cultured on the collagen membrane surface for cytocompatibility assessments using an MTT assay (Figure 5a). More cell proliferation was observed on the surface of collagen membrane samples than on the controls. There was no significant difference among the collagen membranes in metabolic activity. After 77 days of culture, the metabolism of the cells on the collagen membrane increased by 4.5-fold. Cells on the membrane surface exhibited high viability and proliferation (Figure 5b,c), as well as good growth morphology and adherence, as shown in SEM images and live–dead staining images after 7 days.

## 4. Conclusions

A facile and controllable liquid exfoliation method was proposed to directly extract mesoscale collagen fibrils with diameters of 102~159 nm from bovine Achilles tendons in urea/GuHCl. The size of collagen fibrils could be tailored via homogenization treatments. The results of MD simulations and FTIR spectra reveal that the exfoliation of collagen fibrils is attributed to hydrogen bonding and coulomb force interactions with collagen caused by urea and guanidine hydrochloride. The structural features of natural collagen were retained in collagen fibrils. Free-standing collagen membranes were successfully fabricated by applying collagen fibrils as building blocks, exhibiting good transparency, robust mechanical properties, excellent barrier properties, and cytocompatibility. The materials constructed with mesoscopic units showed superior mechanical properties compared with those constructed with macromolecular units. This study provides a method for separating mesoscopic collagen and an approach for processing natural collagen.

## Figures and Tables

**Figure 1 polymers-15-04008-f001:**
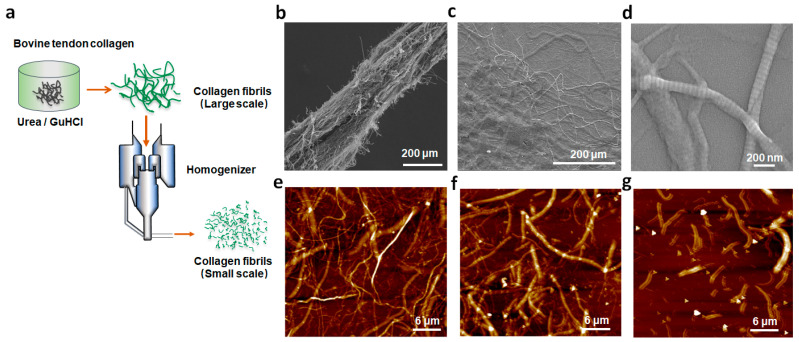
Schematic illustration of the exfoliation of the bovine tendon (**a**); SEM images of collagen fiber bundle from the pre-treated bovine tendon (**b**); disassembled collagen fiber bundle after urea/GuHCl treatment at 70 °C (**c**); and the resulting collagen fibrils after homogenization for 5 times (**d**). AFM images of collagen fibrils after performing homogenization 5 (**e**), 20 (**f**), and 30 (**g**) times.

**Figure 2 polymers-15-04008-f002:**
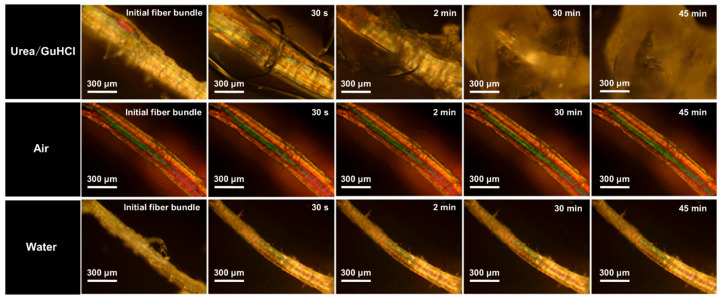
In situ POM presenting collagen fiber bundles in air, water, and urea/GuHCl at 70 °C for different times.

**Figure 3 polymers-15-04008-f003:**
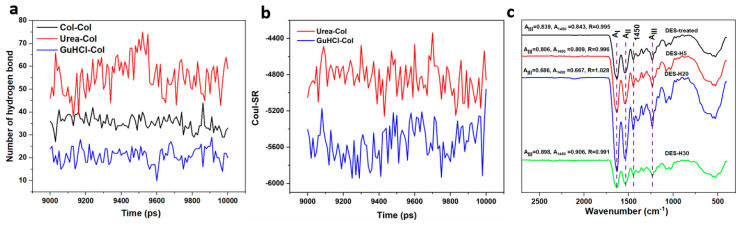
(**a**) Hydrogen bond numbers among urea, GuHCl, and collagen chains; (**b**) Coul−SR of interactions among urea, GuHCl, and collagen chain; (**c**) FTIR spectra of DES-treated collagen without homogenization, and samples after homogenizing DES−H5, DES−H20, and DES−H30.

**Figure 4 polymers-15-04008-f004:**
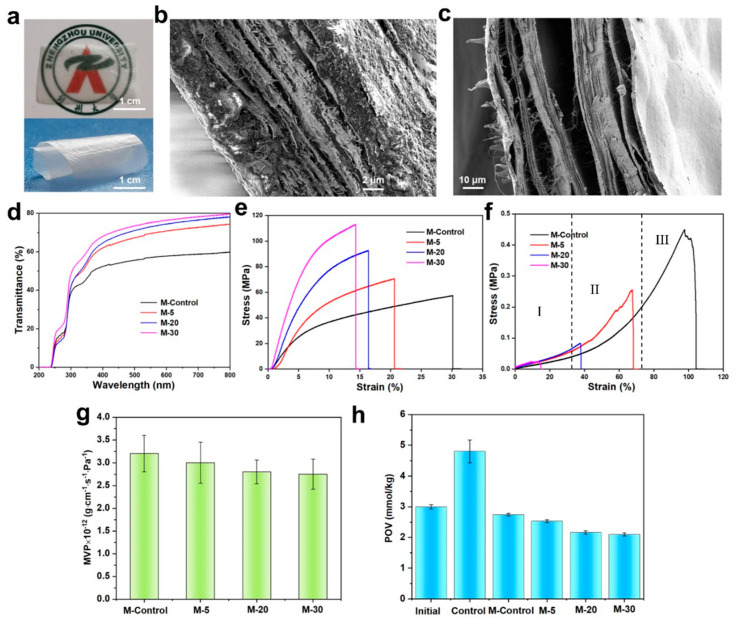
Digital images of spreading and curling M-5 (**a**); SEM images of a cross−section of M-5 in dry (**b**) and wet (**c**) states; transmittance of samples (**d**); stress–strain curves of samples in dry (**e**) and wet state (**f**); WVP (**g**) and POV (**h**) of collagen membranes and control samples.

**Figure 5 polymers-15-04008-f005:**
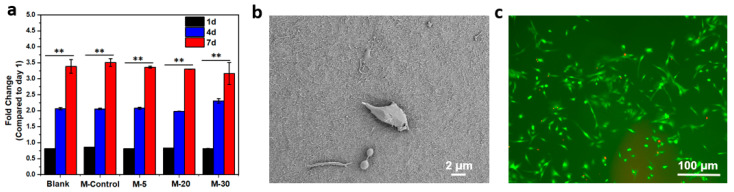
(**a**) Fold change in the metabolic activity for days 1, 4, and 7; results are shown as mean value ± standard deviation, n = 6, ** *p* < 0.1,; (**b**)SEM images show cell morphologies on the surface of M-5 (**c**) and fluorescence images of live (green) and dead (red) L929 fibroblasts on M-5.

## Data Availability

Data will be made available upon request.

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
