# Peer review of "Mesostructured Fibrils Exfoliated in Deep Eutectic Solvent as Building Blocks of Collagen Membranes"

_polymers, 2023, doi:10.3390/polym15194008_

Round 1

Reviewer 1 Report

The authors of the manuscript polymers-2617479 describe the utilization of the deep eutectic solvent (DES) urea/GuHCl  for the exfoliation of the collagen mesostructures. The manuscript needs improvement before publication.

The authors most properly cite and discuss their previous work, where they use the same pattern for the method of preparation of the mesoscale collagen fibrils - Pei, Y., Jordan, K. E., Xiang, N., Parker, R. N., Mu, X., Zhang, L., ... & Kaplan, D. L. (2021). Liquid-exfoliated mesostructured collagen from the bovine achilles tendon as building blocks of collagen membranes. ACS applied materials & interfaces, 13(2), 3186-3198. This previous paper is mentioned only in the Introduction. The advantage of the DES (urea/GuHCl) compared to NaOH/urea must be presented and discussed and separate paragraphs in Discussion section.  My suggestion is to separate Results and Discussion into separate Sections.

Text recycling from authors previous papers (extended verbatim) must be corrected.

…are well organized in native tissues to facilitate remarkable performance and diverse functions. In this work, a facile and controllable liquid  exfoliation method was introduced to directly extract mesoscale collagen fibrils

are organized in native collagen tissues to achieve remarkable and diverse performance and functions. In this work, a facile, low-cost, and controllable liquid exfoliation method was applied to directly extract these collagen mesostructures

In-situ observations under polarizing microscopy (POM) and molecular dynamics simulations revealed the influence of the urea and GuHCl on the tendon collagen. FTIR results confirmed that these fibrils preserved typical structural characteristics of type I collagen. These exfoliated collagen fibrils were then utilized as building blocks to fabricate free-standing collagen membranes, which exhibited good…

in situ observations under polarizing microscopy (POM) and using molecular dynamics simulations revealed the influence of the NaOH/urea system on the tendon collagen. FTIR and XRD results confirmed that these collagen fibrils preserved typical structural characteristics of type I collagen. These isolated collagen fibrils were then utilized as building blocks to fabricate free-standing collagen membranes, which exhibited good

Manuscript polymers-2617479

Pei, Y., Jordan, K. E., Xiang, N., Parker, R. N., Mu, X., Zhang, L., ... & Kaplan, D. L. (2021). Liquid-exfoliated mesostructured collagen from the bovine achilles tendon as building blocks of collagen membranes. ACS applied materials & interfaces, 13(2), 3186-3198.

…collagen-containing wastes and byproducts are generated from the leather and food industries, providing…

…collagen-containing wastes and byproducts are generated from the leather and food industries, providing…

Manuscript polymers-2617479

Chen, J., Liu, J., Yang, W., & Pei, Y. (2023). Collagen and Silk Fibroin as Promising Candidates for Constructing Catalysts. Polymers, 15(2), 375.

The authors must present evidence related to the formation of deep eutectic solvent between urea and Guanidine chloride.

Figure S1 must be presented correctly in the main text – The present mention is ”DES was prepared by 104 mixing GuHCl and urea in the molar ratio of 2:1 at 70℃ until a clear system (Figure S1).” Figure S1 caption is ”Schematic diagram of the experimental process objects.”

Author Response

  1. The authors most properly cite and discuss their previous work, where they use the same pattern for the method of preparation of the mesoscale collagen fibrils - Pei, Y., Jordan, K. E., Xiang, N., Parker, R. N., Mu, X., Zhang, L., ... & Kaplan, D. L. (2021). Liquid-exfoliated mesostructured collagen from the bovine Achilles tendon as building blocks of collagen membranes. ACS applied materials & interfaces, 13(2), 3186-3198. This previous paper is mentioned only in the Introduction. The advantage of the DES (urea/GuHCl) compared to NaOH/urea must be presented and discussed and separate paragraphs in Discussion section. My suggestion is to separate Results and Discussion into separate Sections.

Response: We are very pleased the reviewer felt the review can provide some refreshing insights for our work. The advantage of the DES (urea/GuHCl) compared to NaOH/urea has been presented and discussed in Discussion section. Both urea and guanidine hydrochloride (GuHCl) are able to disrupt the noncovalent bonds (e.g., hydrogen bonds) and decrease the strength of hydrophobic interactions in proteins. DES composed of urea/GuHCl is a promising powerful protein denaturant to directly exfoliate the protein fiber of silk into SNFs by breaking the hydrogen and hydrophobic bonds. Compared with the previous NaOH/urea, a strongly alkaline system, urea/GuHCl provides a milder condition for exfoliation. Moreover, the exfoliation of collagen in NaOH/urea system lasts several hours due to multiple freezing and thawing cycles treatments, while the exfoliation of collagen in urea/GuHCl system requires only 10 min treatment. The above mentioned has been provided in revised manuscript (3.1 ).

According to the reviewer’s suggestion. The results and discussion section has been divided into several separate sections. 3.1 Structure of mesoscale collagen fibrils, 3.2 Analysis of exfoliation process, 3.3 Properties of collagen membranes 

  1. Text recycling from authors previous papers (extended verbatim) must be corrected.

Response: Thank you for your comments. The mentioned text has been corrected in revised manuscript. Please see the following text descriptions.

Text in published paper

Submitted manuscript

Revised manuscript

…are well organized in native tissues to facilitate remarkable performance and diverse functions. In this work, a facile and controllable liquid  exfoliation method was introduced to directly extract mesoscale collagen fibrils

are organized in native collagen tissues to achieve remarkable and diverse performance and functions. In this work, a facile, low-cost, and controllable liquid exfoliation method was applied to directly extract these collagen mesostructures

The mesoscale components of collagen (nanofibrils, fibrils, and fiber bundles) are well organized in native tissues, resulting in superior properties and diverse functions. In this paper, we present a simple and controlled liquid exfoliation method to directly extract medium-sized collagen fibers ranging from 102 to 159 nm in diameter from bovine Achilles tendon using urea/hydrochloric acid and deep eutectic solvent (DES).

In-situ observations under polarizing microscopy (POM) and molecular dynamics simulations revealed the influence of the urea and GuHCl on the tendon collagen. FTIR results confirmed that these fibrils preserved typical structural characteristics of type I collagen. These exfoliated collagen fibrils were then utilized as building blocks to fabricate free-standing collagen membranes, which exhibited good…

in situ observations under polarizing microscopy (POM) and using molecular dynamics simulations revealed the influence of the NaOH/urea system on the tendon collagen. FTIR and XRD results confirmed that these collagen fibrils preserved typical structural characteristics of type I collagen. These isolated collagen fibrils were then utilized as building blocks to fabricate free-standing collagen membranes, which exhibited good

In situ observations under polarized light microscopy (POM) and molecular dynamics simulations revealed the effects of urea and GuHCl on tendon collagen. FTIR study results confirmed that these fibrils retained the typical structural characteristics of type I collagen. These shed collagen fibrils are then used as building blocks to create independent collagen membranes with good and stable mechanical properties, excellent barrier properties and cell compatibility.

Manuscript polymers-2617479

Pei, Y., Jordan, K. E., Xiang, N., Parker, R. N., Mu, X., Zhang, L., ... & Kaplan, D. L. (2021). Liquid-exfoliated mesostructured collagen from the bovine achilles tendon as building blocks of collagen membranes. ACS applied materials & interfaces, 13(2), 3186-3198.

…collagen-containing wastes and byproducts are generated from the leather and food industries, providing…

…collagen-containing wastes and byproducts are generated from the leather and food industries, providing…

The leather and food industries generate large amounts of collagen-containing waste and by-products every year, creating significant opportunities for repurposing natural collagen.

Manuscript polymers-2617479

Chen, J., Liu, J., Yang, W., & Pei, Y. (2023). Collagen and Silk Fibroin as Promising Candidates for Constructing Catalysts. Polymers, 15(2), 375.

  1. The authors must present evidence related to the formation of deep eutectic solvent between urea and Guanidine chloride.

Response: Thank you for your suggestion. Both urea and guanidine hydrochloride (GuHCl) are able to disrupt the noncovalent bonds (e.g., hydrogen bonds) and decrease the strength of hydrophobic interactions in proteins. Urea/GuHCl has been proven to be a deep eutectic solvent and widely used for the treatment of biomass in many previous work [Separation and Purification Technology, 2021, 258,118015][ Green Chem., 2018, 20, 3625][ ACS Sustainable Chem. Eng. 2020, 8, 11, 4499][Biomacromolecules 2020, 21, 4, 1625]. Here, we provide two movies to showing the formation of a deep eutectic solvent by mixing GuHCl and urea in the molar ratio of 2:1, from solid to liquid by heating to 70 ℃, and from liquid to solid by cooling to 25 ℃.

  1. Figure S1 must be presented correctly in the main text – The present mention is ” DES was prepared by 104 mixing GuHCl and urea in the molar ratio of 2:1 at 70℃ until a clear system (Figure S1).” Figure S1 caption is ” Schematic diagram of the experimental process objects.”

Response: Thank you for your comments. We have corrected Figure S1 caption and the present mention in revised text.

Reviewer 2 Report

1- This is an interesting article with moderate novelty in the field. One of the major drawbacks of this article is discussion of polarity of collagen fibrils and how they can contribute in the polarity of oval tissue for mineralization. One interesting article to consider regarding this topic:

Polar nature of biomimetic fluorapatite/gelatin composites: a comparison of bipolar objects and the polar state of natural tissue

Biomacromolecules 16 (9), 2814-2819.   2- Another interesting topic is about how polarity of of collagen can contribute in mineralization of calcium and phosphate ions in tissues. There are several studies related to stimulating gel diffusion method for mineralization of HAp in gelatin as a model study for in vitro HAP/collegen composites. No discussion was represented in this article regarding this method at Introduction section.   3- More input regarding stress-stain curves of samples in dry and wet state should be represented. These curves showed elastic and plastic regions during mechanical test. These regions were not discussed in details. Also the modulus differences of samples in dry and wet state should be more clarifies and compared.

Author Response

1- This is an interesting article with moderate novelty in the field. One of the major drawbacks of this article is discussion of polarity of collagen fibrils and how they can contribute in the polarity of oval tissue for mineralization. One interesting article to consider regarding this topic: Polar nature of biomimetic fluorapatite/gelatin composites: a comparison of bipolar objects and the polar state of natural tissue. Biomacromolecules 16 (9), 2814-2819.

Response: We are very pleased the reviewer felt the review can provide some refreshing insights for our work. The aim of this work is to provide a method for preparing natural mesoscopic collagen and a strategy for preparing collagen materials by using mesoscale collagen as building blocks. However, the in-depth biological application of collagen membrane materials is not involved. We appreciate your suggestion, and cited the above literatures in revised manuscript.

2- Another interesting topic is about how polarity of collagen can contribute in mineralization of calcium and phosphate ions in tissues. There are several studies related to stimulating gel diffusion method for mineralization of HAp in gelatin as a model study for in vitro HAP/collagen composites. No discussion was represented in this article regarding this method at Introduction section.

Response: Thank you for your comments. We have cited some relevant literatures in introduction section. [New J. Chem., 2016, 40, 5495-5500][Journal of Biomedical Materials Research, 2014,102, 5, 1415] [International Journal of Biological Macromolecules, Volume 93, Part B, December 2016, Pages 1390-1401]

3- More input regarding stress-stain curves of samples in dry and wet state should be represented. These curves showed elastic and plastic regions during mechanical test. These regions were not discussed in details. Also the modulus differences of samples in dry and wet state should be more clarifies and compared.

Response: We appreciate your suggestion. Typical stress-strain curves of dry and hydrated membranes were provided in Figure 4 e and 4f. The average data is shown in Table S2 (Supporting materials). The modulus differences of samples in dry and wet state have been presented in revised manuscript. We have provided more discussion regarding the tensile mechanical tests. The dry collagen membranes present a brittle fracture. For hydrated membranes, at lower tensions, the curve has a peak region (I) where the fibrils in the membrane straighten and align along the tension direction. Subsequently, the oriented fibrils deform in the nonlinear region of the curve (stage II) and begin to slide against each other, and the stress-strain behavior is mainly determined by the stretching of the elastic fibrils. The plastic region in which highly oriented fibrils are further deformed, resulting in a significant increase in tensile strength. In stages III, the increase in linear area is attributed to the straightening of kinks among the collagen macromolecules, resulting in reduced disruption of lateral molecular packing in the fibrils. This stretching is also related to the displacement of neighboring collagen molecules.

Table S2 Average mechanical data of dry and hydrated collagen membranes.

Sample

Dry state

Hydrated state

Ultimate stress (MPa)

Ultimate strain (%)

Ultimate stress (KPa)

Ultimate strain (%)

M-5

70.4±0.9

20±7

270.2±14.2

102±9

M-20

87.6±1.7

15±5

80.3±5.3

68±3.2

M-30

112.5±2.5

9±4

27.6±4.4

40±4.9

Control

45.1±1.5

27±3

450±13.2

17±5.1

Round 2

Reviewer 1 Report

The authors responded to the modification requirements. The manuscript is suitable for publication.

Reviewer 2 Report

The article is acceptable now.